# Relict Plants Are Better Able to Adapt to Climate Change: Evidence from Desert Shrub Communities

**DOI:** 10.3390/plants12234065

**Published:** 2023-12-04

**Authors:** Ying Lu, Boran Zhang, Min Zhang, Meiyu Jie, Siqi Guo, Yange Wang

**Affiliations:** School of Architecture, Inner Mongolia University of Technology, Hohhot 010051, China; yinglu_7@163.com (Y.L.); boranzhang0901@163.com (B.Z.); minzhang818@163.com (M.Z.); meiyujie0521@163.com (M.J.); siqiguo2022@163.com (S.G.)

**Keywords:** ancient relict plants, desert ecosystem, climatic change, species distribution model, suitable distribution area

## Abstract

Shrubs are the main dominant plants in arid desert systems and play an important role in maintaining the biodiversity, ecosystem services and stability of desert ecosystems. Studies have shown that the survival of a large number of shrub species in desert areas under the influence of climate change is significantly threatened, with different species showing different response strategies. To test the tolerance of different shrub species to climate change, this study selected 10 dominant shrub species (ancient relict shrub species and regional endemic shrub species) in the Alashan desert area as the research object. Based on a field survey of species distribution, a species distribution model was developed to simulate the suitable distribution area of shrub species under current conditions and under future climate change scenarios. The distribution changes of ancient relict and regional endemic shrub species under the climate change scenarios were tested, and the tolerance of the two types of shrub to climate change was analyzed. The results showed that under different climate change scenarios, except for *Ammopiptanthus mongolicus*, the total suitable area of four out of the five relict plants was relatively stable, the potential distribution area of *Tetraena mongolica* increased, and the future distribution pattern was basically consistent with the current distribution. However, the suitable area of typical desert plants was unstable under different climate change scenarios. Except for *Kalidium foliatum*, the suitable distribution areas of four out of the five shrubs showed different degrees of reduction, and the distribution location showed significant migration. Based on the research results, climate change will lead to the reduction and displacement of the distribution area of typical desert shrubs, while relict shrubs will be less affected by climate change. This is because, compared to desert species, relict plants have a longer evolutionary history and have developed a wider range of adaptations after experiencing dramatic environmental changes. This study provides a scientific basis for actively responding to the impacts of climate change on desert ecosystems.

## 1. Introduction

The impact of climate change on species survival and biodiversity is increasing [1,2,3,4]. The sixth report of the United Nations Intergovernmental Panel on Climate Change (IPCC) states that continued greenhouse gas emissions will lead to a further increase in global temperature. Among the scenarios and model paths considered, the best estimate of global temperature rise is 1.5 ℃ in the near future (2021–2040), and an increase in global temperature rise will lead to multiple hazards [5]. Studies have shown that the risk of extinction will increase with global warming [6,7]. Climate change has an important impact on the distribution of plants, especially endangered species, and the intensification of global climate change poses a serious threat to the maintenance of biodiversity. For example, Parmesan [8] found that climate change increases the risk of extinction for polar and alpine species. Meng et al. [9] found that climate change has a direct impact on the range of the endangered species *Disanthus* endemic in East Asia. Wu et al. [10] found that the spatial distribution pattern of six desert plants, such as *Anabasis brevifolia*, *Haloxylon ammodendron*, *Ephedra przewalskii* and *Ceratoides latens*, changed with climate change. The study by Wan et al. [11] showed that climate change will shift the range of the endangered plant *Taxus cuspidata*. Tao et al. [12] found that the range of the endangered plant *Pinus kwangtungensis* in southern China tends to shift northward with climate change. Therefore, studying the response of different species to climate change and predicting changes in the potential distribution of species under climate change scenarios are crucial for the conservation and utilization of endangered plants [13].

Different species have different response strategies and adaptabilities to climate change. In the process of long-term adaptation to harsh habitats, relict plants have formed different adaptation mechanisms and survival strategies. For example, the study of Qin et al. showed that the distribution area of the relict plant *Potaninia mongolica* will increase in the future under the climate change scenario, and precipitation was the main limiting factor dictating its distribution [14]. The study of *Tsuga longibracteata* by Tan et al. showed that the precipitation in the driest month was the dominant climatic factor affecting its distribution, and its future distribution pattern did not change significantly under the influence of climate change, showing the ‘refuge in situ’ characteristic [15]. A study of *Cathaya argyrophylla* by Ran et al. found that the potential distribution area of this species will expand in the future [16]. Zhao et al. found that precipitation was the dominant factor controlling the distribution of *Gymnocarpos przewalskii*, and the suitable distribution area of this species increased under different climate change scenarios [17]. A study of the relict plant *Liriodendron chinense* by Zhai et al. found that the wettest season rainfall, driest season rainfall, and monthly mean temperature difference between day and night were the main limiting factors affecting its distribution [18]. The geometric center of its suitable distribution area remained unchanged under the influence of climate change. The tolerance of relict plants to climate change is strong, the suitable distribution area simulated under different climate scenarios fluctuates little or increases, and temperature is not the main distribution-limiting factor.

At present, studies on Tertiary relict plants mainly focus on community structure, population maintenance mechanism, population genetics and genetic geography, but less on plant origin and evolutionary history. Shang et al. [19] investigated six species of relict plants, i.e., *Cyclocarya paliurus*, *Nyssa sinensis*, *Liquidambar acalycina*, *Liquidambar formosana*, *Emmenopterys henryi* and *Euptelea pleiospermum*, and analyzed their population structure and regeneration strategies. Tang et al. and He et al. investigated *Metasequoia glyptostroboides* [20], *Ginkgo biloba* [21], *Liriodendron chinense* [22], *Tetracentron sinense* [23,24] and *Taiwania cryptomerioides* [25] distributed in China, and the results showed that the unstable habitat disturbed to a certain extent was beneficial to the population maintenance of Tertiary relict plants and occupied the greatest dominance. Compared with typical desert plants, the relict plants have a longer evolutionary history. Therefore, we speculate that the relict plants have developed different survival strategies in the process of tolerating harsh habitats for a long time and have evolved a wider range of adaptations after undergoing dramatic environmental changes, which may be more vital than we thought [26].

The Alashan desert area is located in the Eurasian hinterland, far from the ocean, which is a typical mid-temperate arid area [27]. The East Alashan–West Erdos region is one of the most concentrated distribution areas of endemic plants and psammophytes in the northwestern arid region of China [28,29], where there are many Tertiary relict plants that have existed since the Early Tertiary in the northern hemisphere [30,31]. From the early Oligocene, due to the cold climate, a large number of plants distributed in the middle and high latitudes and around the north gradually retreated to the low latitudes [32], and during the Late Tertiary to Quaternary, they retreated southward to the three major glacial refugia of East Asia, North America, and Southwest Europe [30,33,34]. The special ecological background of Alashan desert makes it the preferable area of the plant endemic phenomenon in arid and semi-arid areas of China, and it is the distribution gathering place of endemic plant genera in Inner Mongolia Plateau and Central Asia (eastern Central Asia) [35]. Its plant composition is dominated by xerophytes, super-xerophytes and semi-shrubs, and there are few perennial herbs and legumes. The dominant shrubs are Chenopodiaceae, Rosaceae, Leguminosae and Compositae, forming a unique vegetation landscape in the desert [27]. The ecological environment in this region is harsh, and the area suitable for human production and living is only 6%, with drought and water scarcity and sparse vegetation [36]. It has experienced many frequent climate change events [37,38,39], intense climate change events [40,41,42] and high temperature events. The extreme minimum temperature was −34.4 °C (Norrigong, 24 January 2008), and the extreme maximum temperature was 44.5 °C (Guaizi Lake, 28 July 2020). The coldest month is January, with an average temperature ranging from −10.9 to −7.5 °C, and the warmest month is July, with an average temperature ranging from 23.8 to 28.4 °C. The average annual precipitation is 39.3~224.2 mm [43]. According to the data of Alashan Meteorological Observatory, from 1962 to 2021, the annual average number of sandstorm days was 10.7. The natural ecosystem in this region has limited adaptability and is more vulnerable to serious or even irreversible damage [36]. Based on this, this study selected the dominant shrub species in the Alashan desert area as the research object, simulated the potential distribution areas of relict shrub species and typical desert shrub species under different climate change scenarios, and compared and analyzed the adaptability of the two types of shrubs to climate change to test the hypothesis that relict plants can better adapt to climate change.

## 2. Materials and Methods

### 2.1. Study Area and Species Distribution Survey

The study area is located in the eastern part of the temperate arid desert subregion of the Hexi Corridor–Alashan Plateau in China. To facilitate mapping, Alashan City, Inner Mongolia Autonomous Region, is used as the main study area (Figure 1). Alashan is bordered by Gansu Province to the west, Bayannur City to the northeast and Wuhai City to the southeast. Alashanzuoqi county is located in the southeast of Alashan, Alashanyouqi county is located in the center of Alashan, and Ejinaqi county is located in the northwest of Alashan, which is also the westernmost tip of Inner Mongolia Autonomous Region. The total area of the region is 270,000 km^2^, of which the desert area is 63,700 km^2^ [43].

The study area is located in the hinterland of the Asian continent, which has a typical continental climate. It is dry and rainless, windy and sandy, cold in winter and hot in summer. The climate characteristics of the four seasons are distinct, and the temperature difference between day and night is large. Affected by the southeast monsoon, the rainy season is mostly concentrated in July, August and September; the precipitation decreases from the southeast to the northwest and evaporation decreases from southeast to northwest [43]. In the northern part of the study area, westerly winds prevail, and in the south, southeasterly winds prevail.

Due to the unique climatic and geomorphological conditions in the Alashan desert area, a large number of rare and endangered species are present. There are 913 species of wild plants and 6 wild shrub nature reserves in the study area [44]. The main protected species include endangered shrubs *Tetraena mongolica*, *Ammopiptanthus mongolicus*, *Amygdalus mongolica* and *Sarcozygium xanthoxylon*. In this study, 10 dominant shrub species were selected for research, as shown in Table 1.

*Nitraria tangutorum* is a deciduous dwarf shrub. Zhang et al. [45] considered that *Nitraria* first appeared between the Cretaceous and the Early Paleocene and hypothesized an Early Paleogene (65 Mya) origin of *Nitraria* based on molecular data. Amber Woutersen et al. [46] found the oldest fossil pollen grain of *Nitraria*, at least 53 Myr old, and dated it to the Paleogene based on its characteristics. *Reaumuria songarica* is an ultra-xerophyte with strong stress resistance [47]. According to paleomagnetic measurements and fossil evidence, it underwent the desertification of Asia that began at least 22 million years ago [48]. *Tetraena mongolica* is an ancient Mediterranean relict plant from 140 million years ago [28,49]. *Sarcozygium xanthoxylon* is a succulent xerophytic deciduous shrub [50]. Bellstedt et al. [51] found that Asian *Zygophyllum* and *T. mongolica* originated independently in Africa and may be of Eocene and Miocene age, respectively. Wu et al. [52] showed that Asian *Zygophyllum* originated in the early Oligocene (30.39 Ma, HPD%: 21.53–39.81 Ma) and indicated that Asian *Zygophyllum* differentiated in the early Miocene (19.56 Ma, 95% HPD: 11.25–28.78 Ma). *Ammopiptanthus mongolicus* is an evergreen broad-leaved leguminous shrub [53]. According to the historical data [54], Liu et al. [55] concluded that *Ammopiptanthus* is one of the species of *Trib. Podalyrieal* evolved to adapt to different habitats at the end of the Miocene–Eocene. *Amygdalus mongolica* is a xerophytic deciduous shrub. Zhao [56] showed that *A. mongolica* often grew along the surface with good hydrothermal conditions, forming a “green corridor” in the desert area. *Halocnemum strobilaceum* is a component of the Mediterranean–Central Asian flora [57], and the *H. strobilaceum* community is the most widespread plant community in the halophytic desert [58]. *Kalidium foliatum* community is another widespread halophyte community, and its habitat is similar to that of *H. strobilaceum* community, but its salt tolerance is less [58]. *Convolvulus tragacanthoides* is a desert salt–alkaline plant, and it is found on all continents except circumpolar areas [59,60,61]. *Artemisia ordosica* is a major indicator plant of desert grassland desertification in the northern and northwestern temperate regions of China [62].

The desert area was divided into 30 × 30 km grids, the position of shrub species in the grid was investigated and the species distribution map was drawn for the construction of a species distribution model. At the same time, the spatial point pattern of shrubs in different shrub communities was investigated for the analysis of the relationship among shrub species.

**Table 1 plants-12-04065-t001:** Overview of shrub species.

	Family	Species	Habitat and Distribution	Ecological Traits
relict plants	Zygophyllaceae	*Sarcozygium xanthoxylon*	Deserts, steppe deserts and desertified grassland areas	Drought resistance
	Nitrariaceae	*Nitraria tangutorum*	Desert and semidesert lake basin sands	Salt-tolerant, sand fixation
	Tamaricaceae	*Reaumuria songarica*	Deserts and desert steppe zones	Drought resistance, salt tolerance, sand collection
	Zygophyllaceae	*Tetraena mongolica*	Steppe desert, Yellow River terrace, steppe desert area	Drought tolerance
	Leguminosae	*Ammopiptanthus mongolicus*	Sandy and gravelly textures in desert areas	Super xerophytic structure, strong stress resistance
typical desert plants	Rosaceae	*Amygdalus mongolica*	Gobi Desert Region in Central Asia	Drought tolerance, cold hardiness
	Chenopodiaceae	*Kalidium foliatum*	Wet, fluffy saline soils of deserts, desert steppes and grasslands in Eurasia	Salt tolerance
	Compositae	*Artemisia ordosica*	Sandy areas of steppe, desert steppe to steppe deserts	Wind erosion resistance, sand burial resistance
	Chenopodiaceae	*Halocnemum strobilaceum*	Southern Europe, Western and Northern Asia and Northern Africa	Salt tolerance
	Convolvulaceae	*Convolv* *u* *lus tragacanthoides*	Dry slopes in semidesert areas and mountain basins	Drought resistance

Relict plants are plants with a long origin, many related groups are extinct, relatively isolated, and slow evolution. Typical desert plants are plants that can survive in desert conditions. According to the “Flora of China”, this study refers to *Zygophyllum* and *Sarcozygium* as *Zygophyllum*. The above sources are [63,64,65,66,67,68].

### 2.2. Data Analysis

#### 2.2.1. Species Distribution Model

(1)Abiotic factors

Nonbiological factors are mainly divided into soil attribute factors, bioclimatic factors and topographic factors. Soil attribute data were derived from the World Soil Database (http://www.fao.org/, accessed on 9 September 2022). Bioclimatic data were derived from the WorldClim (https://www.worldclim.org/, accessed on 23 September 2022) data website, and 19 bioclimatic variables covered by the data were used as biometeorological factors. Topographic data were obtained from the WorldClim website using SRTM DEM digital elevation data to generate bioclimatic data. Nonbiological data were resampled, unified with species distribution data, and converted into ASC files for model construction. The nonbiological factors selected are detailed in Table 2.

(2)Biological factors.

In addition to the influence of environmental factors, species growth is also affected by the distribution of other species. Therefore, it is necessary to analyze the interaction between species and obtain the species types that affect the distribution of the species to be predicted. The spatial point pattern analysis method was used in this study. Based on the data of shrub species distribution points in different communities, the R package spatstat was used to calculate Ripley’s K_(12)_ function [69]. Based on 100 Monte Carlo simulations, the confidence interval was calculated, and the relationship between different shrub species was analyzed. The analysis results are shown in Figure 2.

The Maxent model was used to simulate and predict the potential distribution area of shrub species in the Alashan desert under different climate change scenarios. Seventy percent of the training samples were randomly selected, and thirty percent of the distribution data were selected as the prediction samples [70,71,72]. The receiver operating characteristic (ROC) curve was used as the measurement method of model simulation accuracy, and the area under the curve (AUC) enclosed by the ROC curve and abscissa was used as the evaluation index. The AUC values were all greater than 0.9, indicating that the model had good fit.

#### 2.2.2. Climate Change Scenario

The setting of climate change scenarios is an important basis for the analysis of the potential impacts of future climate change. The climate change data were set as proposed in the CMIP6 model in the Sixth Assessment Report of the IPCC (AR6) [73,74]. Based on the CMIP5 climate change scenario, the Shared Socio-economic Path (SSP) was added as the climate change scenario. SSP refers to the scenario model of global climate change without the influence of relevant climate policies [75]. The set of SSP climate scenarios includes SSP126, SSP245, SSP370, and SSP585.

The SSP126 scenario predicts the future under the green growth paradigm [76], which represents an environmentally friendly sustainable development scenario and low greenhouse gas emissions, with the radiative forcing reaching 2.6 W·m^−2^ in 2100 [77], the total amount of agricultural land being greatly reduced and the forest area increasing [78,79]. SSP245 is a low stability scenario, which represents the intermediate scenario of social and economic development and a medium level of greenhouse gas emissions, with the radiative forcing reaching 4.5 W·m^−2^ before 2100 and not exceeding this value [80]. SSP370 represents the regional competitive development path of the medium–high forcing scenario, which represents the strong expansion of global cropland and pastureland, which increase by 40% and 7%, respectively, from 2010 to 2100, leading to large-scale deforestation, and the radiative forcing would reach 7.0 W·m^−2^ in 2100 [80]. In the SSP585 scenario, fossil fuel use is projected to double global food demand and greenhouse gas emissions are projected to triple during this century [80], with radiative forcing reaching 8.5 W·m^−2^ in 2100 [77].

Climate change scenarios were obtained from WorldClim. WorldClim provided the 20-year averages of bioclimatic data under the SSP126, SSP245, SSP370 and SSP585 global climate change scenarios from 2041 to 2060 (2050s) [75,81,82,83]. The climate change scenario data were resampled and unified with species distribution data and converted into ASC files for model construction.

## 3. Results

### 3.1. Shrub Species Distribution

Based on the species data from the field investigation, a distribution point map of shrub species in the Alashan desert area was drawn (Figure 3).

The range of distribution points of different species is quite different, but most of them are concentrated in the southeast of the study area, which is related to the characteristics of the species themselves. In the field investigation, it was found that the relict species *N. tangutorum* was concentrated on flat terrain with more sediment content but was distributed throughout the whole Alashan desert area. *S. xanthoxylon* was concentrated in the area with more sediment content, and it was distributed in a relatively flat terrain. *R. songarica* was mainly distributed in the valley area [84], in the eastern part of the Alashan desert area and in the southeastern part of the study area. *T. mongolica*, as an endangered protected species, had few habitats and few individuals [85], and was distributed on a small-scale in the western, eastern and northern regions of the Alashan desert. *A. mongolicus* was distributed in the eastern part of Alashan desert area. This shrub had a small range, which was found in barren hills and rocky Gobi besides sandy gravel soil [53].

The typical desert plant *A. mongolica* was mainly distributed in high mountains and steep rock walls and was sporadically distributed in the eastern part of the Alashan desert with high aggregation intensity [86]. *K. foliatum* was concentrated in saline–alkali land and was distributed within a small range in the west, east and north of the Alashan desert area. *A. ordosica* was distributed more commonly along rivers and within a small range in the southeast, west and north of the Alashan desert area. The distribution range of *H. strobilaceum* was small, mainly in the eastern part of the Alashan desert area; *C. tragacanthoides* was distributed in the east and south of the Alashan desert.

### 3.2. Prediction of Shrub Species Distribution under Different Climate Change Scenarios

The Maxent model was used to simulate and predict the suitable distribution area of shrub species in the current Alashan desert area, as shown in Figure 4.

The suitable distribution area of species predicted via simulation basically covered the effective distribution points of all species obtained from the investigation, indicating that the suitable distribution area of each species simulated under the current climatic conditions was basically consistent with its actual distribution range.

The results showed that relict plants were mainly distributed in the southeast of the study area, and there was also a small distribution in the north–central part of the study area. The specific distribution of each species was as follows:(1)The potential distribution areas of the relict plants *N. tangutorum*, *S. xanthoxylon* and *R. songarica* were mainly in the southeastern part of the study area, among which *N. tangutorum* had a higher probability of distribution in the eastern part of the study area. The overall suitable area of the *S. xanthoxylon* was biased toward the middle of the study area; the suitable distribution area of *R. songarica* in the southern part of the study area was large, and it was similar to the suitable area of *S. xanthoxylon*, which was related to the interaction between the species.(2)The endangered relict plant *T. mongolica* was mainly distributed in the east and north of the study area, and the distribution probability in the north was low.(3)The endangered relict plant *A. mongolicus* was mainly distributed in the central and eastern parts of the study area, and the distribution probability in the central part was low.

Typical desert plants were widely distributed in the eastern part of the study area. The specific distribution of each species was as follows:(1)The typical desert plant *A. mongolica* was concentrated in the eastern part of the study area, and very few areas showed a high probability distribution.(2)*H. strobilaceum* and *K. foliatum* were distributed in the southeastern part of the study area, and the distribution area was scattered.(3)The suitable distribution area of *A. ordosica* was large, and it was often distributed in the eastern boundary of the study area.(4)The suitable distribution area of *C. tragacanthoides* was located in the southeast of the study area, and the distribution probability was high in the south.

The Maxent model was used to simulate and predict the suitable distribution areas of shrub species in the Alashan desert under four climate change scenarios in the coming years until 2050. The results are shown in Figure 5.

Under different climate change scenarios, the suitable distribution areas of relict plants were basically consistent with the current distribution forms of all species, mainly concentrated in the southeast of the study area. The distribution of each species was as follows:(1)The suitable distribution areas of the relict plants *S. xanthoxylon*, *N. tangutorum* and *R. songarica* under the four climate change scenarios were all concentrated in the southeastern part of the study area, which was consistent with the current distribution pattern of each species, but there was a trend toward a shift to the southern part of the study area.(2)The suitable distribution area of the relict plant *T. mongolica* was located in the north and east of the study area, and the suitable area in the north was facing the risk of contraction.(3)The suitable area of the relict plant *A. mongolicus* was located in the middle and east of the study area. The distribution probability in the middle was low, and the distribution patterns under the four climate scenarios were basically the same.

Under different climate change scenarios, the suitable distribution areas of typical desert plants had changed greatly. The distribution of each species was as follows:(1)The suitable distribution area of *A. mongolica*, a typical desert plant, changed significantly under the four climate change scenarios. The suitable distribution areas under the SSP126, SSP370 and SSP585 climate change scenarios were located in the northern and eastern parts of the study area. The distribution area in the northern part of the study area disappeared under the SSP245 climate change scenario.(2)The suitable distribution areas of *H. strobilaceum* and *K. foliatum* were located in the southeastern part of the study area.(3)The probability of distribution of *C. tragacanthoides* in the eastern part of the study area was higher.(4)The suitable distribution area of *A. ordosica* was located in the middle and south of the study area.

### 3.3. Differences in Tolerance of Different Shrub Species to Climate Change

Based on the prediction map of the suitable distribution area of 10 shrub species under four climate change scenarios generated using the Maxent model above, the proportion of area change of relict plants (Table 3 and Figure 6) and typical desert plants (Table 4 and Figure 7) was calculated.

The results showed that there were differences in the suitable distribution area changes of endangered relict shrub species and typical desert shrub species under different future climate scenarios: (1) Among the relict plants, except for *A. mongolicus*, the suitable area of most plants was relatively stable. The suitable area of *A. mongolicus* always decreased, and the decrease rate was stable at approximately 13%. The species for which the suitable distribution area always increased were *R. songarica* and *T. mongolica*. The average increase in the suitable distribution area of *R. songarica* was approximately 4.5%, and the change range was small. The increase in the suitable distribution area of *T. mongolica* under the four climate scenarios was 6–19%. The total suitable distribution areas of *R. songarica*, *S. xanthoxylon* and *N. tangutorum* did not change much. (2) The suitable area of typical desert plants was unstable. Except for *K. foliatum*, the suitable distribution area of most species showed a decreasing trend to varying degrees. The suitable area of *K. foliatum* always increased, and its total area was the largest under the SSP585 climate change scenario, reaching 10,227 km^2^, and the smallest was under the SSP245 climate change scenario at 9956 km^2^. The suitable area of *C. tragacanthoides* always decreased, and the decrease ranged from 2% to 17%. The total distribution area of *H. strobilaceum* was similar under the four climate change scenarios, and it was slightly larger than 7454 km^2^ under the SSP370 climate change scenario and slightly smaller than 7054 km^2^ under the SSP126 climate change scenario. The change in the different distribution probability areas of the species was offset, resulting in little change in the total suitable area. The total suitable areas of *A. mongolica* and *A. ordosica* changed greatly under the different climate change scenarios. The total suitable area for *A. mongolica* under all climate scenarios except for the SSP370 scenario, under which a small increase was observed, and the change ratio ranged from 1% to 10%. The total suitable distribution area of *A. ordosica* was 29,650 km^2^ under the SSP585 climate change scenario and 32,172 km^2^ under the SSP370 climate change scenario. Compared with the current suitable area, there was great uncertainty in the area increases and decreases.

## 4. Discussion

### 4.1. Climate Change Is Not the Main Limiting Factor for the Distribution of Relict Plants

Among the nonbiological factors examined in this study, the soil attribute factor is one of the main factors limiting plant growth, and the biometeorological factor is the main factor limiting the establishment and development of species niches, especially at medium and large scales [87]. The main factor affecting plant growth in desert areas is water, and soil moisture is the main source of water for plant growth. In the dry season, water deficit is prone to occur. To adapt to arid environments, vegetation will form a spatial pattern.

In this study, the potential distribution areas of 10 shrubs reflected well the close relationship between geographical distribution and environment, but the environmental factors limiting the distribution of relict plants and typical desert plants were different. (1) The response of the two groups of plants to different soil properties was different: relict plants such as *S. xanthoxylon*, *N. tangutorum* and *T. mongolica* were concentrated in areas with high sand content, species such as *K. foliatum* were concentrated in saline–alkali land and species such as *A. ordosica* were more frequently distributed along rivers. (2) The distribution of shrub species was restricted by topographic factors, and the factors of elevation, slope and aspect had great influence on shrub species. Relict plants such as *S. xanthoxylon* and *N. tangutorum* were mainly distributed in relatively flat terrain, the typical desert plant *A. mongolica* was mainly distributed in hilltop areas with larger slopes and *K. foliatum* was mainly distributed in valley areas.

The distribution of plant populations is usually the result of the interaction of ecological and historical geographical factors. There are a large number of relict plants in the study area, which have adapted to arid environments in the long-term ecological evolution process. The suitable distribution area of the relict plant *A. mongolicus* predicted by the model was partly located on the eastern edge of the Ulan Buh desert, and a small part was distributed along the northern part of the Helan Mountains. The simulation results confirmed that the places where *A. mongolicus* occurred were areas with good water and heat conditions in the local environment, and it could not tolerate excessively harsh arid environmental conditions [88]. This was related to its physiological and ecological characteristics and explained why it does not conform to the hypothesis that “relict plants can better adapt to climate change”. Among the typical desert plants, except for *A. ordosica*, the suitable distribution areas of other shrubs were mainly concentrated in the southeastern part of the study area because this area had the most suitable combination of water and heat conditions compared with other desert areas. Typical desert plants are generally significantly affected by temperature and precipitation, and temperature is not the main limiting factor for the distribution of relict plants.

### 4.2. There Is a General Positive Correlation between Relict Shrub Species

The suitable distribution areas of some relict shrub species were very similar. The analysis showed the following relationships among these species: (1) Although there was no correlation between species, the response of the species to the environment was similar. For example, *S. xanthoxylon* and *N. tangutorum* were concentrated in sandy land and relatively flat terrain, so these species were distributed at the edge of the desert. (2) There were interactions between species. The interspecific relationship between *S. xanthoxylon* and *R. songarica* showed a significant positive correlation, and the suitable areas of the two plants in the eastern part of the study area were similar. Interspecific relationships have an important influence on environmental adaptation. The above situation showed that the suitable distribution area of the species was affected not only by the environment but also by the interspecific relationship.

### 4.3. Relict Shrub Species Have Developed Rich Ecological Strategies in Their Long Evolutionary History

Except for *A. mongolicus* and *T. mongolica*, the distributions of the remaining species under different future climate change scenarios were consistent with the current suitable distribution area, and the distribution range showed no considerable change. As super-xerophytic shrubs in desert areas, *A. mongolicus* and *T. mongolica* have a strong ability to adapt to arid environments. Due to the influence of extreme climate, the plant community structure was singular, the ecosystem was unstable, and the habitat suitability of the community decreased. *A. mongolicus* has a limited geographical distribution and isolated taxonomic status [89]. To maintain the integrity of leaf structure and leaf function in harsh environments, its foliar nutrient demand is higher than that of other plants, resulting in other relict plant populations having stronger environmental adaptability than *A. mongolicus* under different climate change scenarios. The distribution of *T. mongolica* was characterized by a long and narrow gap, and temperature had a significant influence this species. A previous study showed that during the growth of *T. mongolica*, the harsher the environment, the stronger the reproductive capacity [90], which also explains why the suitable area of *T. mongolica* always increased. The results of this study showed that the suitable range of *T. mongolica* has a tendency to migrate to high latitudes to meet its growth needs under human intervention.

Relict plants and typical desert plants have different strategies to adapt to the environment. Most relict plants, such as *S. xanthoxylon*, *N. tangutorum* and *R. songarica*, are important constructive species in desert areas. During the long evolutionary process, these species have interacted with habitats and gradually developed many internal physiological and external morphological adaptation strategies to promote their growth and development, which is conducive to the survival and expansion of these species in desert areas. The physiological metabolic rates of the photosynthesis, respiration and transpiration of most relict plants are lower than those of typical desert plants under the same conditions. This is the result of the long-term evolution of adaptation strategies to arid climates and environments. Gao et al. [91] found that the branches of *Nitraria* were often specialized as thorns to reduce the transpiration area, and *Tetraena* formed fleshy short rod-shaped leaves. The leaves of *Zygophyllum* were not only fleshy, but the whole plant was often laid flat on the ground to prevent water transpiration and strong light burns. The leaves of *Nitraria* were grayish white, which had a certain reflective effect on strong sunlight. The cuticle on the leaf surface of *Zygophyllum* was thick or slightly leathery and shiny to prevent the mesophyll cells from being burned. To make efficient use of water, the leaves of *Zygophyllum* and *Tetraena* were apparently fleshy, and the spongy tissue cells of the mesophyll specialized as large water-storing parenchyma cells that stored a lot of water. Compared with other genera, the transport tissue of *Zygophyllum* was more developed and the transport efficiency was higher. Meng et al. [92] found that the adaptation of shrubs such as *A. mongolicus* and *R. songarica* to the environment was mainly as follows: the leaves were membranous leaves, fleshy leaves and acupuncture leaves. The fruits were mainly dried fruits. The water ecotype was mainly xerophyte, and the water use efficiency of seeds was diverse.

However, while adapting to the growth environment of arid areas, relict plants also show the problems of slow growth and difficult recovery after being destroyed. For example, the relict plant *T. mongolica* is often in an unfavorable competitive position in the community after entering the adult stage, and it tends to give way to other strong xerophytic shrubs such as *S. xanthoxylon* and *N. tangutorum*. Field observations showed that the patches formed after landscape fragmentation were gradually occupied by the close relatives of *N. tangutorum* and *T. mongolica*. *A. mongolica*, a typical desert plant, is a dominant species in the Alashan desert area. Its interspecific competition is weak, the endocarp is hard and thick and mature seeds cannot be sowed far, so it presents an aggregated distribution pattern. *A. mongolica* generally cannot form a large desert community and can only be distributed in mountains and valleys to form a local fragment community. This fragmentated distribution is likely to affect the genetic diversity and evolutionary potential of *A. mongolica*, resulting in the disappearance of its suitable area in the northern part of the study area in the simulation results.

After decades of evolution, relict plants have strong environmental adaptability, are less affected by climate change and have a small range of suitable distribution areas. However, the study found that the optimal distribution area of endangered relict plant populations such as *T. mongolica* and *A. mongolicus* was increasingly narrow. It is speculated that island isolation may occur in the future, and problems such as difficulty in gene exchange and heritability decline may occur. These plants generally lack the ability to regenerate naturally and coupled with the vulnerability of the distribution environment and the intensification of climate change, there is a problem of limited regeneration [85,93,94,95].

This study is based on the niche of relict plants captured in the field survey, which has certain limitations, and some speculations were not confirmed. In the future, we will strengthen research on the functional characteristics of these shrub species and learn more about the environmental adaptation and ecological strategies of these relict plants. When analyzing the driving factors of plant population distribution, environmental variables should be fully considered to further explore the influence of abiotic factors on plant population distribution patterns. Due to the limitation of experimental conditions, it is difficult to confirm the existence of allelopathy among species. In the future, distinguishing differentiation and interspecific competition through controlled laboratory experiments will help to better understand the ecological adaptation mechanism of plants.

The protection and rational utilization of relict plant resources play a very important role in the ecological protection and improvement of desert areas. It is recommended that nature reserve planning and biodiversity conservation should take into account the adaptation of plants to climate change.

## Figures and Tables

**Figure 1 plants-12-04065-f001:**
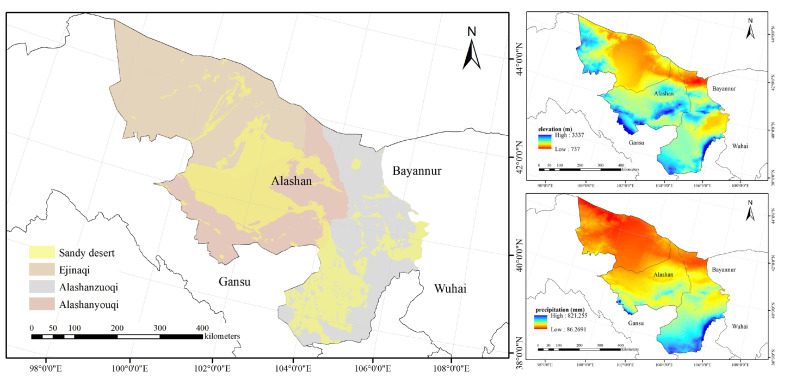
Overview of the study area (the yellow area represents a sandy desert with almost no vegetation. The topography data adopted SRTM DEM digital elevation data obtained from the WorldClim website. The precipitation data were derived from https://www.worldclim.org/, accessed on 23 September 2022, which was version 2.1 of WorldClim, released in 2020).

**Figure 2 plants-12-04065-f002:**
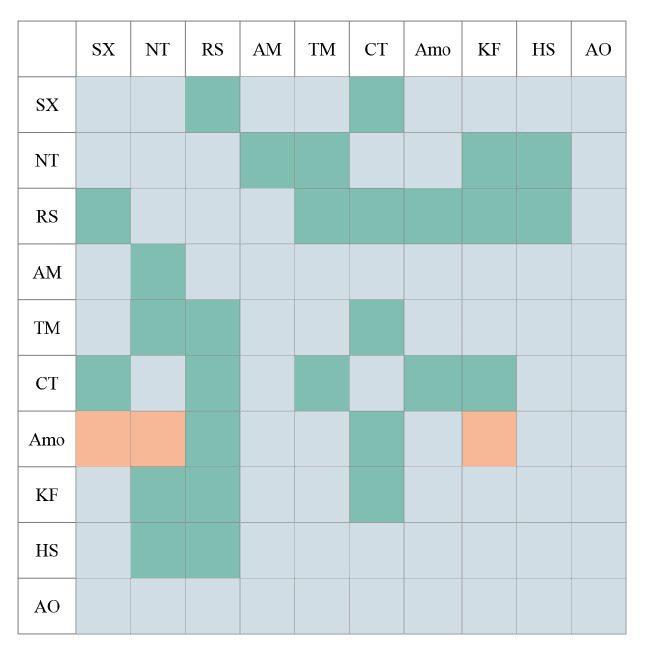
Interspecies relationship diagram. Legend: SX, *Sarcozygium xanthoxylon*; NT, *Nitraria tangutorum*; RS, *Reaumuria songarica*; AM, *Ammopiptanthus mongolicus*; TM, *Tetraena mongolica*; CT, *Convolvulus tragacanthoides*; Amo, *Amygdalus mongolica*; KF, *Kalidium foliatum*; HS, *Halocnemum strobilaceum*; AO, *Artemisia ordosica*. Green indicates a positive correlation, orange indicates a negative correlation and gray indicates a nonsignificant correlation.

**Figure 3 plants-12-04065-f003:**
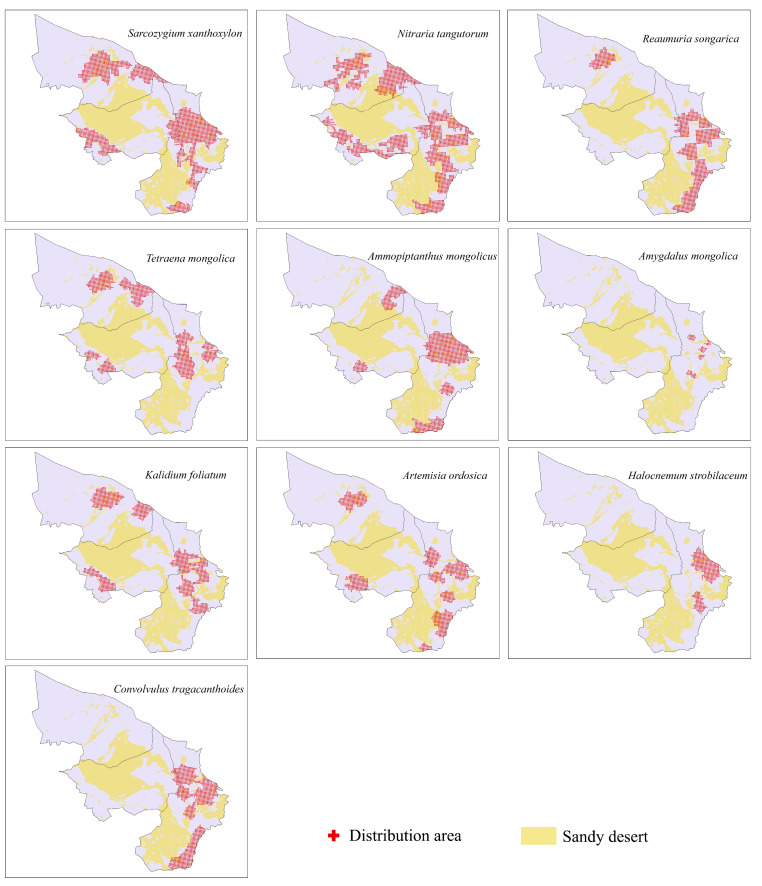
Distribution points of shrub species.

**Figure 4 plants-12-04065-f004:**
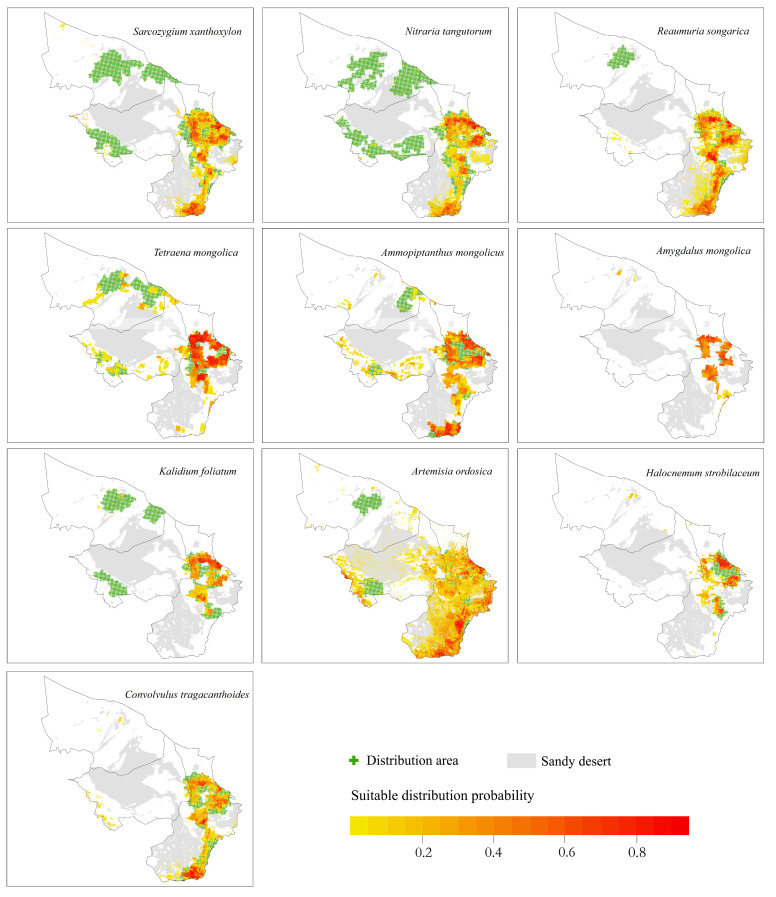
Probability plot of the suitable distribution of shrub species.

**Figure 5 plants-12-04065-f005:**
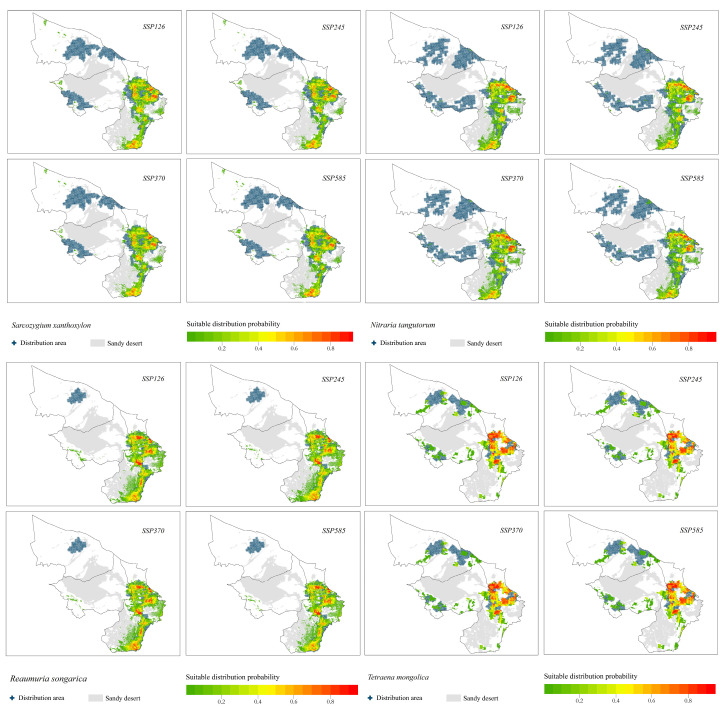
Probability plot of the suitability distribution of 10 shrub species under different climate change scenarios.

**Figure 6 plants-12-04065-f006:**
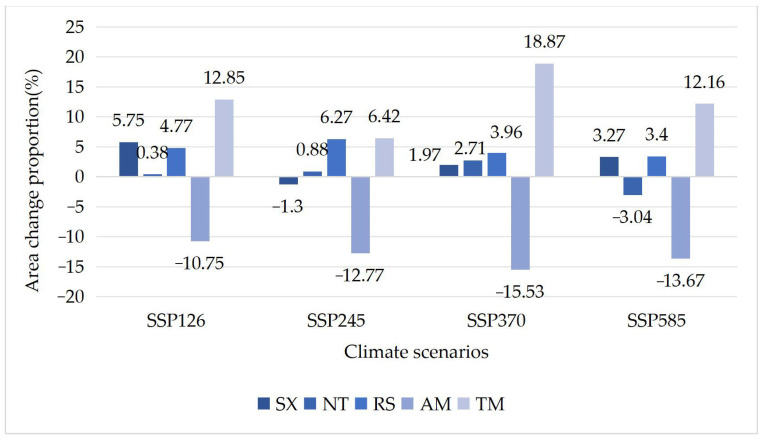
The change in suitable range for relict plants compared to the current climate. The “−” is the percentage reduction in range.

**Figure 7 plants-12-04065-f007:**
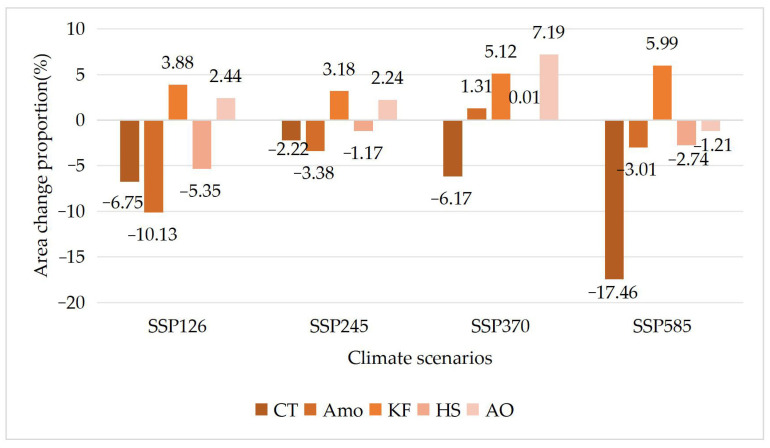
Change in suitable range for typical desert plants compared to current climate. The “−” is the percentage reduction in range.

**Table 2 plants-12-04065-t002:** Description of abiotic factors.

Soil Property Factors	Topographic Factors	Biometeorological Factors	
Silt concentration	Slope	Average annual temperature	Average temperature of the warmest quarter
Silt content	Aspect of slope	Daily difference in mean temperature	Average temperature of the coldest quarter
Clay content	Altitude	Isothermality	Annual rainfall
Organic carbon content		Coefficient of seasonal variation of temperature	Precipitation in the wettest month
PH		Maximum temperature in the hottest month	Precipitation in the driest month
Carbonate or lime content		Minimum temperature in the coldest month	Seasonal variations in precipitation
Sulfate content		Annual difference in temperature	Precipitation in the driest quarter
		Average temperature of the wettest quarter	Wettest quarter precipitation
		Average temperature of the driest quarter	Warmest quarter precipitation
			Coldest quarter precipitation

**Table 3 plants-12-04065-t003:** Suitable areas for relict plants under current and four future climate scenarios.

Species	Current Area (km^2^)	Area under SSP126 Scenario (km^2^)	Area under SSP245 Scenario (km^2^)	Area under SSP370 Scenario (km^2^)	Area under SSP585 Scenario (km^2^)
SX	16,343	17,283	16,130	16,665	16,878
NT	14,760	14,816	14,947	15,217	14,366
RS	17,986	18,843	19,113	18,699	18,598
AM	21,554	21,276	20,064	22,410	21,145
TM	18,853	19,237	18,802	18,207	18,608

**Table 4 plants-12-04065-t004:** Suitable areas of typical desert plants under current and four future climate scenarios.

Species	Current Area (km^2^)	Area under SSP126 Scenario (km^2^)	Area under SSP245 Scenario (km^2^)	Area under SSP370 Scenario (km^2^)	Area under SSP585 Scenario (km^2^)
CT	12,311	11,480	12,038	11,552	10,161
Amo	12,857	11,555	12,423	13,025	12,470
KF	9649	10,023	9956	10,143	10,227
HS	7453	7054	7366	7454	7249
AO	30,014	30,748	30,686	32,172	29,650

## Data Availability

All the data are available from the corresponding author upon reasonable request.

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
