# Peer review of "Relict Plants Are Better Able to Adapt to Climate Change: Evidence from Desert Shrub Communities"

_plants, 2023, doi:10.3390/plants12234065_

Round 1
Reviewer 1 Report
Comments and Suggestions for Authors
Notes on review of PLANT paper
This is good work under difficult conditions in a remote and harsh environment of Alashan. The authors are to be commended.
Some concerns
The shrubs etc that make up the bulk of the biomass in these arid areas have quite long life spans (60 years +).Recruitment occurs rarely because there is no need to introduce competitors for water and nutrients in to the parent plant’s space. Close examination of the size, stem diameter etc will show how there are cohorts and “waves” of recruitment - and then only in a circumstance where there is “A co-occurrence of favourable events”. This means that seeds can germinate and seedlings can establish only rarely. Such favourable combinations of rainfall, temperature etc only occur every 15-20 years. We therefore see even-aged stands of key woody species. I say all of this because I cannot understand how any inference can be drawn about “migration, or extending the geographical range of a species.
A lot of your analysis is highly speculative. Climate change in such a highly environment as Alashan cannot be detected in a period that is so short. Year to year variations occur and even within seasons there may be big differences from one year to the next. Heavy frost or heatwave, torrential rains and drought sequences all play their part in determining whether there will be a successful recruitment to the population of the key plants.
In lines 189 and following to line 225 we see statements about present-day ‘presence/absence of key species’ but no indication about their relative age and density. All were recruited to the population before the measurements to detect temperature increases were even thought of. Some plants could be 50-60 years of age.
The results from work on mapping the distribution pattern of key relict species is valuable in itself and worth publishing. We know little about the lifecycle, longevity, seed germination conditions etc. (the autecology of these relict species) so it is difficult to make too much of present-day distribution.
Please refer to the attached sheet of comments (by line number)
Items requiring attention or explanation
Line 52, 54 60 62 109-117 there is no authority show after the species name but elsewhere you show some e.g. in the Tables but not everywhere. Please be consistent. Also muse italics for all botanical names unless you are referring to a well known species e.g. Artemisia sp. that has the same common name as its botanical name.
BE CONSISTENT Use botanical names in italics (with authority) Tables and text should use same schema
124 So too Convolvolus
158, 226 200-202 reference is made to MAXENT but may give false outcomes
189 you refer to clumps and remark on distribution of plants within the community. Maybe the concept of ‘nearest neighbor distance’ could be used to explain tolerance “clumping” in some species.
237 confusion between temperature variation and attribution to global warming
303 migration is hard to prove. We know so little about the cohorts of existing plants, their ages etc
335 reference is made to allelopathy and whilst that is possible there is no evidence to support the idea.
356 ability to migrate depends on reproductive strategy, clonal trees like Aspen or seed dependent like Populus euphratica Seed dispersal mechanisms vary and hard seededness ensures an available seed bank that can allow recruitment to the population when the co-occurrence of favourable events comes along (say every 15-20 years). The life strategy of these relicts is to ensure that minimum number of mature plants survive and seed occasionally.
369 regeneration and rejuvenation see earlier notes on recruitment, even-aged stands and recruitment of new individuals
383 “After ‘years’ should be ‘decades’ or ‘centuries’
389 Speculation
References
No. 40 sums it up “so many variables…”

Reviewer 2 Report
Comments and Suggestions for Authors
My main problem with the manuscript is in its underlying basic assumptions and definitions, as provided in L. 50-51, 64-68: Relict species do not form a group of species defined by certain traits, but are simply an assortment of unrelated species that occur outside their chohrotypes or the biomes in which they evolve. While this may be an indication of their ability to withstand a broad range of environmental stresses (implying adaptation), they may simply be on the road to extinction (implying no adaptation). Relict species are not necessarily better adapted to harsh conditions (a problematic term on its own) and have not necessarily adapted to such condition during their evolution. Sometimes, or some would claim most times, they survive in more favorable niches that conserve the (less harsh) conditions in which they evolved. Other times, their populations are declining.
Another important issue is that the Discussion is highly speculative, and is not really based on results (or, at least, not results that are shown). In the Results section, there is no reference to environmental variables, species traits and their relationships with distribution patterns.
The last sentence of the manuscript does not really say anything. The authors simply claim their study is important, but do not show how and why in previous sections (i.e., do not lead the reader to this conclusion).
L. 10: “Ecosystem services” and “ecosystem functioning” are two separate concepts.
L. 14: What does “main” mean? Dominant? Widely distributed?
L. 19: I think “adaptability” is too strong a word, considering that there is no measure of fitness related to any directly applied stressor.
L. 19-23, 23-26: would prefer “four out of the five species’ over “most”.
L. 28: Do you mean that desert shrubs do not “interact with desert conditions” throughout their evolution? It would seem more logical that the relicts are more generalist (have survived thus far both desert and non-desert conditions) whereas desert species are more specialized and restricted to desert conditions.
L. 37-48: This entire paragraph is a collection of over-generalized statements. You should provide examples or explanations (mechanisms).
L. 69-77: What are you trying to say here? Please consider my comment on L. 28 above.
L. 78: What is a “typical desert”? What is the “enrichment zone of Tertiary relict species”? Define species traits and evolutionary histories, not their current distribution.
L. 82: Some ecologists claim that Mediterranean species are actually less susceptible to climate change, because they are used to seasonal and decadal fluctuations (e.g., droughts). There are many papers about this from Israel, for example.
L. 83: What is “harsh”? Provide variables and values.
Fig. 1: Some terms in the graphical legend are not explained in the text. There are also no climate variables (e.g., rainfall or SPEI) indicated in the map.
L. 112-117: Organize.
Table 1: Which species are “relict” and which are “typical”? Why is “Genus name” a column? The title “protection level” is incorrect, and the information in this column is unclear and inconsistent.
Table 2: For soil, you provide only codes and not explanations. For topography, you provide explanations but not codes. For biometeorological factors you provide both. I would discard the codes altogether.
L. 167-172: Describe shortly each scenario, using bioclimatic variables.
Fig. 3: It seems as if all the species you studied occur only outside the desert, including “typical desert species”.
L. 203-206: Showing simulated and actual distribution on the same panel will show this the best.
L. 207-223, 232-249: Can you summarize according to your grouping? “Typical desert plants are distributed mainly… whereas relict species are distributed mainly…”
Fig. 5: No normal person can read this figure. It is too small and does not clearly indicate and separate scenarios and species groups.
L. 250: With no actual measure of “adaptation” (fitness relative to environmental variable), I would not use this word.
Tables 3-4: Consider converting this into a figure. It will be far more straightforward and intelligible.
Round 2
Reviewer 2 Report
Comments and Suggestions for Authors
Thank you for the time you took to consider my comments and revise the manuscript. Nevertheless, I find the revision unsatisfactory. I list the issues in order of importance (most important first):
Issue 1:
Response 1 (and 8):
While your hypothesis may be valid, you need to track evolutionary history in order to test it. Otherwise, it remains a possible (hypothesised) explanation for your results, rather than a tested hypothesis.
Response 2:
Same problem – you try to explain differences in species distribution without understanding the species. This is not botany/ecology, but seeking hypothetical explanation for geostatistics.
Issue 2:
Response 10:
You consistently confuse “life history” with “evolutionary history”. These are two completely different things!
L. 85-86: Strong claims require strong evidence.
L. 86-87: Apart from the problematic terminology – your hypothesis, although reasonable, is very difficult to accept. Evolution and adaptation do not necessarily make species more flexible or generalist, maybe even vice-versa. Also note that adaptability itself is a trait (i.e., adaptation).
Issue 3:
Despite the disclaimers/caveats in its ending, the Discussion remains mostly speculative. You show a pattern worth investigating, but not discuss it based on evidence.
Issue 4:
L. 161-162: Please provide references for each species, including numbers showing a longer evolutionary history as a species/genus.
Issue 5:
Response 6:
I am not sure if “tolerance” is correct, since the term refers to ability to maintain performance under certain conditions. You should spell out that you tested their distribution changes following climate change scenarios.
Minor issues:
L. 48: Put the citation after the author’s name, not at the end of the sentence. I would also avoid stating he/she “believed”, assuming they present evidence.
Fig. 1: Please clarify that the yellow is a sandy desert devoid of almost any vegetation.
Table 1: The “genus” column is redundant.
Fig. 3-5: The best thing would be, for each panel, to overlay the map in Fig. 3 over the maps in Fig. 4-5, or any other strategy that shows how the range changes, instead of many maps among which the reader should compare.
Comments on the Quality of English LanguageEnglish is mostly fine. Any minor issue can be dealt with during proofing.
Round 3
Reviewer 2 Report
Comments and Suggestions for Authors
Overall, I am happy with the revisions, and as you shall see my comments now are mostly technical.
L. 13: Change “adaptability” to “tolerance”
L. 24: “A few” or just one?
L. 30-34: This is not a sentence
L. 51: “increases” (climate change is still going on)
L. 53: “in East Asia”
L. 88-92: It would be better to indicate which reference is for which species, e.g., “In China, population of relict species Species One [1], Species Two [2-3], … are maintained …”
L. 118-119: “preferable”, not “most obvious”
L. 121-126: English issues
L. 130: C*d is not a common temperature variable to use, what about daily means of the warmest and coldest months? Also, use “mean annual rainfall”.
L. 187-188: Do not use initials before mentioning the full name, and if you mention family – do this for all species. This entire section, while generally good, is long and uneven. Consider adding as much of the information in Table 1 instead of the body text.
L. 270-276: Please indicate what each scenario is in terms of temperature, rainfall, etc.
L. 567-595: While tis part is important and in the correct place, it is written in a manner that makes the readers feel like you did only half the job and more like a research proposal. It can be written in a more positive/charismatic voice.
Comments on the Quality of English LanguageSome parts, especially revisions, require moderate editing.
